# Augmentation of vaccine-induced humoral and cellular immunity by a physical radiofrequency adjuvant

Yan Cao[1], Xiaoyue Zhu[1], Md Nazir Hossen[1], Prateek Kakar[1], Yiwen Zhao[1] & Xinyuan Chen[1]

Protein/subunit vaccines often require external adjuvants to induce protective immunity. Due to the safety concern of chemical adjuvants, physical adjuvants were recently explored to boost vaccination. Physical adjuvants use physical energies rather than chemicals to stimulate tissue stress and endogenous danger signal release to boost vaccination. Here we present the safety and potency of non-invasive radiofrequency treatment to boost intradermal vaccination in murine models. We show non-invasive radiofrequency can increase protein antigen-induced humoral and cellular immune responses with adjuvant effects comparable to widely used chemical adjuvants. Radiofrequency adjuvant can also safely boost pandemic 2009 H1N1 influenza vaccination with adjuvant effects comparable to MF59-like AddaVax adjuvant. We find radiofrequency adjuvant induces heat shock protein 70 (HSP70) release and activates MyD88 to mediate the adjuvant effects. Physical radiofrequency can potentially be a safe and potent adjuvant to augment protein/subunit vaccine-induced humoral and cellular immune responses.

[1] Biomedical and Pharmaceutical Sciences, College of Pharmacy, University of Rhode Island, 7 Greenhouse Road, Avedisian Hall, Room 480, Kingston, RI 02881, USA. Correspondence and requests for materials should be addressed to X.C. (email: xchen14@uri.edu)

Traditional live-attenuated and inactivated whole-cell vaccines have good immunogenicity. Yet, live-attenuated vaccines face risks to cause diseases in immunocompromised populations and inactivated whole-cell vaccines are associated with high reactogenicity. Protein/subunit vaccines are gaining increasing popularity due to improved safety. Yet, protein/subunit vaccines have low immunogenicity and often require addition of external adjuvants to induce protective immunity[1]. Adjuvants come from the Latin word adjuvare, meaning to help. Adjuvants potentiate vaccine-induced immune responses and can be used to enhance vaccine effectiveness, stimulate specific immune responses, elicit broad protection, and reduce antigen amounts and vaccine doses[2]. Adjuvants are increasingly used in modern vaccines and are sometimes essential for success of vaccines.

Adjuvant development in the past mainly relied on empirical experience and only a few adjuvants have been approved for human use[2,3]. Alum adjuvant has been the most widely used adjuvant in the globe since its first empirical use in 1930s[4]. Alum adjuvant induces Th2-biased immune responses with major enhancement on humoral immunity[2,4]. MF59 is a squalene emulsion-based adjuvant first approved to boost seasonal influenza vaccine efficacy in 1997[5]. MF59 has since been approved in more than 30 countries with over 100 million MF59-incorporated influenza vaccine doses distributed[6]. MF59 also induces Th2-biased immune responses[7,8]. Besides Alum and MF59, several Adjuvant Systems AS01 (MPL and QS21 in liposome), AS03 (another squalene emulsion-based adjuvant), and AS04 (MPL adsorbed on Alum) have been approved to boost malaria RTS,S, 2009 H1N1 pandemic influenza, and human papillomavirus vaccine efficacy, respectively[9,10]. Besides approved adjuvants, a number of experimental adjuvants also exist, like complete and incomplete Freund's adjuvants, most of the pathogen-associated molecular patterns (PAMPs), and cytokines[2,11,12]. Yet, experimental adjuvants tend to induce significant local and/or systemic adverse reactions that preclude their use in prophylactic vaccines in humans[2].

Due to the safety concern of chemical adjuvants, physical adjuvants have recently been explored to boost vaccination without introducing potentially harmful chemicals into the body. Physical adjuvants use physical energies rather than chemicals to stimulate tissue stress to enhance vaccine-induced immune responses. Dr. Matzinger proposed danger model in 1994 to explain adaptive immunity could be driven by tissue stress, which led to identification of different endogenous danger signals, such as heat shock proteins (HSPs), uric acid, adenosine triphosphate (ATP), double-strand DNA[13,14]. Endogenous danger signals are a group of molecules that are sequestered from recognition by innate immune systems under normal conditions and can release under tissue stress to alert innate immune systems[13,14]. Endogenous danger signals may serve as vaccine adjuvants to boost vaccine-induced immune responses. In fact, non-PAMP-based Alum and MF59 adjuvants have been found to induce uric acid and ATP release, respectively, to at least partially mediate their adjuvant effects[8,15]. Considering physical energies can be briefly applied to induce tissue stress without long-lasting effects, physical adjuvants are less likely to induce persistent local or systemic adverse reactions. Physical adjuvants can also conveniently boost vaccination without modification of vaccine manufacturing.

Different lasers have been explored to boost intradermal (ID) vaccination. Non-invasive green and near-infrared lasers were found to enhance antigen uptake, maturation, and mobility of skin antigen-presenting cells to boost ID vaccine-induced immune responses[16–21]. Non-ablative fractional laser (NAFL) was used to generate micro-sterile inflammation arrays and recruit plasmacytoid dendritic cells (pDCs) to boost ID

vaccination[22,23]. Besides laser, low-frequency ultrasound was found to activate epidermal Langerhans cells to boost transcutaneous immunization[24]. Adjuvant effects of near-infrared laser were found to be comparable or slightly better than Alum adjuvant in boosting influenza vaccination[16,19]. NAFL can be combined with topical Imiquimod (IMIQ) adjuvant to significantly boost influenza vaccination with NAFL/IMIQ adjuvant effects comparable to MF59-like AddaVax adjuvant[22]. Although most of the physical adjuvants reported to date still remain in the preclinical stages, the safety and potency of physical adjuvants prompted us to explore whether other types of physical energies could induce similar or better adjuvant effects by stimulation of different types of tissue stress and innate immune responses.

Radiofrequencies (RFs) are alternating electromagnetic waves. RFs at medium-high frequencies (0.3-10 MHz) generate tissue heating with broad applications in Aesthetics (e.g., skin tightening) and Medicine (e.g., tumor ablation)[25,26]. This study explores cosmetic RF treatment of a small area of the skin ($2 \times 2$ cm$^2$) followed by ID delivery of protein/subunit vaccines into RF-treated skin to induce tissue stress and boost ID vaccine-induced immune responses in murine models. We find non-invasive RF treatment induces local inflammation, enhances antigen uptake and maturation of dendritic cells (DCs) in the skin and draining lymph nodes (dLNs), and significantly augments ID ovalbumin (OVA) and recombinant hemagglutinin (rHA)-induced humoral and cellular immune responses with RF adjuvant (RFA) effects non-inferior to widely used chemical adjuvants. RFA can also safely boost pandemic 2009 H1N1 influenza (pdm09) vaccination with adjuvant effects comparable or superior to MF59-like AddaVax adjuvant based on vaccine doses. RFA is further found to stimulate heat shock protein 70 (HSP70) release and activate myeloid differentiation primary response 88 (MyD88) to mediate its adjuvant effects.

## Results

**Non-invasive RF induces low-level local inflammation.** A cosmetic fractional bipolar RF device equipped with an electrode array for delivery of RF energies into the skin was used to evaluate its potential adjuvant effects in the absence of tissue damage. Our pilot studies found RF treatment of murine dorsal skin at high-energy setting for more than 2 min could induce instant skin damage, while no skin damage could be found if treatment was 1.5 min or less. At the end of 1.5 min of RF treatment, skin turned white with clear signs of electrode array positioning on the skin surface (Supplementary Fig. 1a). Skin then gradually returned to its normal color and morphology within 20 min (Supplementary Fig. 1a). Skin histological analysis found 1.5 min of RF treatment induced immune cell recruitment 2 h after treatment (upper panels, Supplementary Fig. 1b). RF also increased dermal collagen levels 24 h after treatment (lower panels, Supplementary Fig. 1b), indicative of induction of thermal stress and neo-collagen synthesis, in line with function of the cosmetic RF device. Due to the induction of significant tissue stress without tissue damage, RF treatment at high-energy setting for 1.5 min was selected for further evaluation of its impacts on innate and adaptive immunity in the following studies.

RF-induced local inflammation, such as cytokine/chemokine release and immune cell recruitment, was explored and compared with that induced by widely used chemical adjuvants, such as Alum, MPL, and AddaVax. As shown in Fig. 1a, RF induced relatively low levels of cytokine gene expression except *IL6* as compared with chemical adjuvants (Fig. 1a). *IL6* expression was significantly induced by RF, Alum, and AddaVax, and more significantly induced by MPL (Fig. 1a). RF also induced relatively low levels of chemokine gene expression as compared to chemical

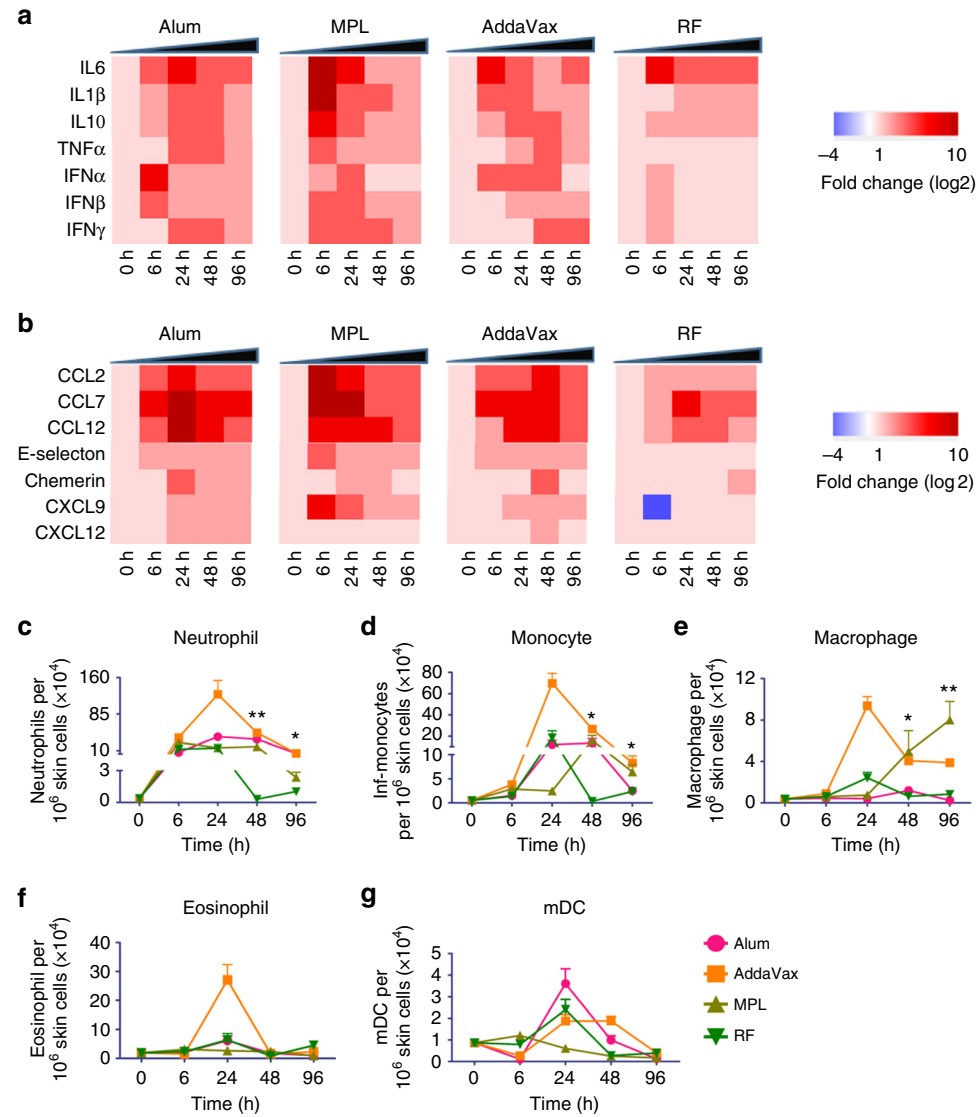

**Fig. 1** RF induces low-level local inflammation. Dorsal skin of C57BL/6 mice were exposed to RF or intradermally injected with 20 µl Alum (1:1 volume ratio in PBS), AddaVax (50%, vol/vol in PBS), or MPL (25 µg). Adjuvant-treated and non-treated skins were dissected at indicated times. **a**, **b** Heat map of relative cytokine (**a**) and chemokine (**b**) gene expression. Total RNA was extracted followed by reverse transcription and real-time PCR analysis of cytokine and chemokine gene expression using *GAPDH* as an internal control. The baseline gene expression level was set at 1. **c**–**g** Different innate immune cell levels in RF and adjuvant-treated skin. Skin was digested in collagenase D and dispase to prepare single-cell suspensions. Cells were then stained with fluorescence-conjugated antibodies followed by flow cytometry analysis of levels of different cell types: neutrophils (**c**), monocytes (**d**), macrophages (**e**), eosinophils (**f**), and mDCs (**g**) (Supplementary Fig. 2). $n = 4$. Student's $t$-test was used to compare differences between groups at 48 and 96 h. *, $p < 0.05$; **, $p < 0.01$

adjuvants (Fig. 1b). RF induced minimal expression of leukocyte attractant *E-selectin* and lymphocyte attractant *CXCL9* and *CXCL12*, slight expression of DC/macrophage attractant *Chemerin* and monocyte chemoattractant *CCL2*, and significant expression of monocyte chemoattractant *CCL7* and monocyte/eosinophil chemoattractant *CCL12* (Fig. 1b). In comparison, chemical adjuvants induced vigorous expression of *CCL2*, *CCL7*, *CCL12*, and significant expression of *E-selectin*, *Chemerin*, *CXCL9*, and *CXCL12* (Fig. 1b).

RF-induced immune cell recruitment was then explored and compared with that induced by chemical adjuvants. As shown in Fig. 1c–f, AddaVax induced the most significant recruitment of innate immune cells (neutrophils, monocytes, macrophages, eosinophils) that peaked at 24 h. Alum and MPL also induced significant recruitment of innate immune cells and most of the innate immune cells peaked at 24 or 48 h (Fig. 1c–f). RF induced

significant recruitment of innate immune cells that peaked at 6 or 24 h and then returned to baseline levels at 48 h (Fig. 1c–f). Skin neutrophil, monocyte, and macrophage levels were significantly higher at 48 and 96 h in at least two out of three chemical adjuvant groups than that in RF group (Fig. 1c–e). These data indicated that RF treatment induced more transient immune cell recruitment than chemical adjuvants. Noticeably, skin myeloid DC (mDC) levels were significantly increased after RF, Alum, and AddaVax treatment, and to a lesser degree after MPL treatment (Fig. 1g).

**RF increases function of specific DC subsets**. DCs play crucial roles in bridging innate and adaptive immunity. Next we explored whether RF treatment could improve function of DCs. We first used CD11c as a pan-DC marker and found more DCs took up

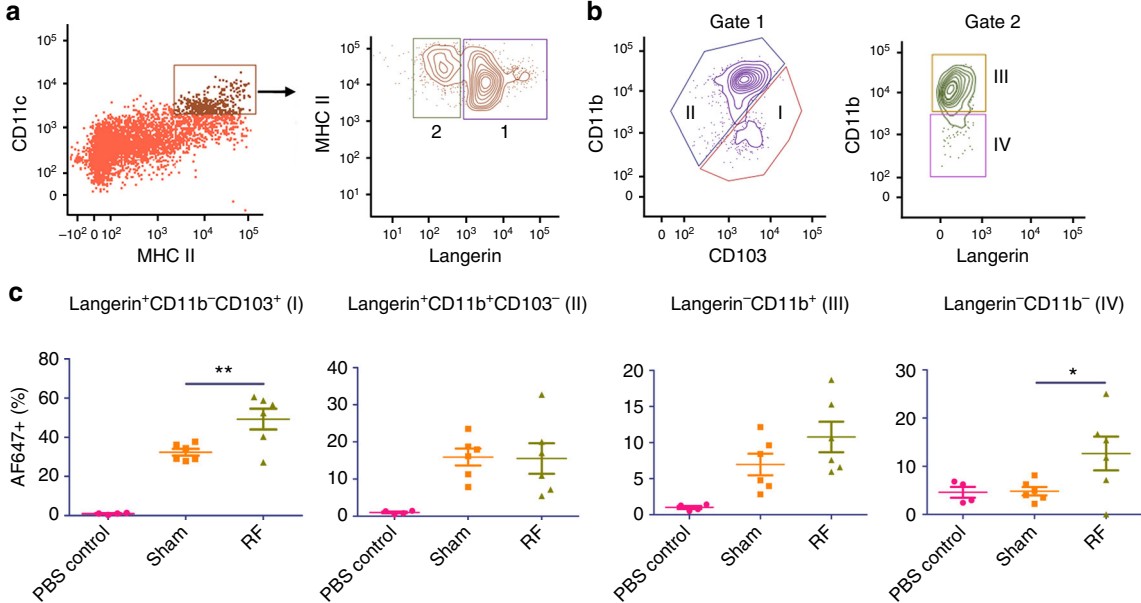

**Fig. 2** RF increases antigen uptake of DCs in skin. Lateral back skin of C57BL/6 mice was exposed to RF or sham treatment followed by ID injection of 2 μg AF647-OVA into RF or sham-treated skin. ID injection of PBS served as control. Skin was harvested 24 h later and digested in collagenase D and dispase to prepare single-cell suspensions. Skin cells were then stained with fluorescence-conjugated antibodies followed by flow cytometry analysis. **a** Live cells were gated and plotted based on CD11c and MHC II expression. CD11c$^+$MHC II$^+$ cells were gated and plotted based on Langerin expression. **b** Langerin$^+$ cells (gate 1) were plotted based on CD11b and CD103 expression into Langerin$^+$CD11b$^-$CD103$^+$ (I) and Langerin$^+$CD11b$^+$CD103$^-$ subsets (II), whereas Langerin$^-$ cells (gate 2) were plotted based on CD11b expression into Langerin$^-$CD11b$^+$ (III) and Langerin$^-$CD11b$^-$ subsets (IV). **c** Percentage of AF647$^+$ cells was analyzed for each DC subset. $n = 4$ for PBS control and 6 for sham and RF groups. Student's $t$-test was used to compare differences between RF and sham groups. *$p < 0.05$; **$p < 0.01$. Representative of two independent experiments

fluorescent AF647-OVA following ID injection into RF-treated skin as compared to sham-treated skin (48% vs. 30%) (Supplementary Fig. 3a, b). In addition, mean fluorescence intensity (MFI) of AF647 in AF647$^+$ DCs was significantly increased in RF group as compared with that in sham group (12,110 vs. 7712) (Supplementary Fig. 3c). In dLNs, more AF647$^+$ DCs were found in RF group than that in sham group (Supplementary Fig. 3d). In addition, MFI of costimulatory marker CD80 and CD40 in AF647$^+$ DCs was significantly higher in RF group than that in sham group (Supplementary Fig.3e–g).

Due to the existence of different DC subsets in skin and dLNs, we further explored whether RF specifically acted on certain DC subsets. Skin CD11c$^+$MHC II$^+$ cells can be divided into four major subsets based on expression of Langerin, CD11b, and CD103: epidermal and dermal Langerin$^+$CD11b$^+$CD103$^-$ cells, dermal Langerin$^+$CD11b$^-$CD103$^+$, dermal Langerin$^-$CD11b$^+$, and dermal Langerin$^-$CD11b$^-$ cells (Fig. 2a, b)[27]. We found RF treatment significantly increased antigen uptake in dermal Langerin$^+$CD11b$^-$CD103$^+$ and dermal Langerin$^-$CD11b$^-$ DCs (Fig. 2c). Antigen uptake in dermal Langerin$^-$CD11b$^+$ DCs was also increased after RF treatment, but the difference between RF and sham groups failed to reach a statistically significant level ($p = 0.0782$, Fig. 2c). Different DC subsets also exist in dLNs, including classical DCs (cDCs), migratory DCs (migDCs), and pDCs, which can be differentiated based on the relative expression of CD11c and major histocompatibility complex (MHC) II (Fig. 3a)[28]. RF had no significant impact on cDC, migDC, and pDC levels in dLNs (upper panels, Fig. 3b). Yet, RF significantly increased antigen uptake in all DC subsets (middle panels, Fig. 3b) and surface expression of CD80 in cDC and pDC subsets (lower panels, Fig. 3b). Considering cDC can be further divided into CD11b$^+$ and CD103$^+$ subsets (Fig. 3c) and migDC can be further divided into Langerin$^+$, Langerin$^-$CD11b$^-$, and

Langerin$^-$CD11b$^+$ subsets (Supplementary Fig. 4a)[18,27], we further explored impacts of RF treatment on individual cDC and migDC subsets. As shown in Fig. 3d, RF significantly increased percentage of CD11b$^+$ cDC as well as antigen uptake and surface expression of CD80 in CD11b$^+$ cDC. As for migDCs, RF significantly increased antigen uptake but not cell percentage or surface expression of CD80 in migDC subsets (Supplementary Fig. 4b). These results indicated that RF treatment increased antigen uptake and/or maturation of specific DC subsets in skin and dLNs.

**RFA augments OVA-induced humoral and cellular immunity.** Model antigen OVA was then used to explore RFA effects. RFA was further compared with MF59-like AddaVax adjuvant to boost ID OVA immunization. After prime/boost immunization, RFA and AddaVax were found to increase anti-OVA antibody titer by ~ 15- and 24-folds, respectively (Fig. 4a). No significant difference in anti-OVA antibody titer was found between RFA and AddaVax groups ($p > 0.05$, Fig. 4a). Induction of OVA-specific CD8$^+$ T-cell responses were evaluated by H-2K$^b$-restricted OVA$_{257-264}$ (SIINFEKL) tetramer staining of peripheral blood mononuclear cells (PBMCs) 1 week after boost. As shown in Fig. 4b and 4c, RFA more significantly increased tetramer$^+$ CD8$^+$ T cells than AddaVax. Mice were further challenged with E.G7-OVA. We found tumor growth was significantly inhibited in RFA group but not in AddaVax group (Fig. 4d). Tumor volume on day 22 and 24 was significantly smaller in RFA group than that in AddaVax group (Fig. 4d). All mice succumbed to tumor within 2 weeks in non-immunized or no adjuvant group, whereas 20% mice in AddaVax group and 40% mice in RFA group were tumor-free within the study period (Fig. 4e). These results indicated that RFA was more potent than AddaVax to stimulate OVA-specific CD8$^+$ T-cell responses.

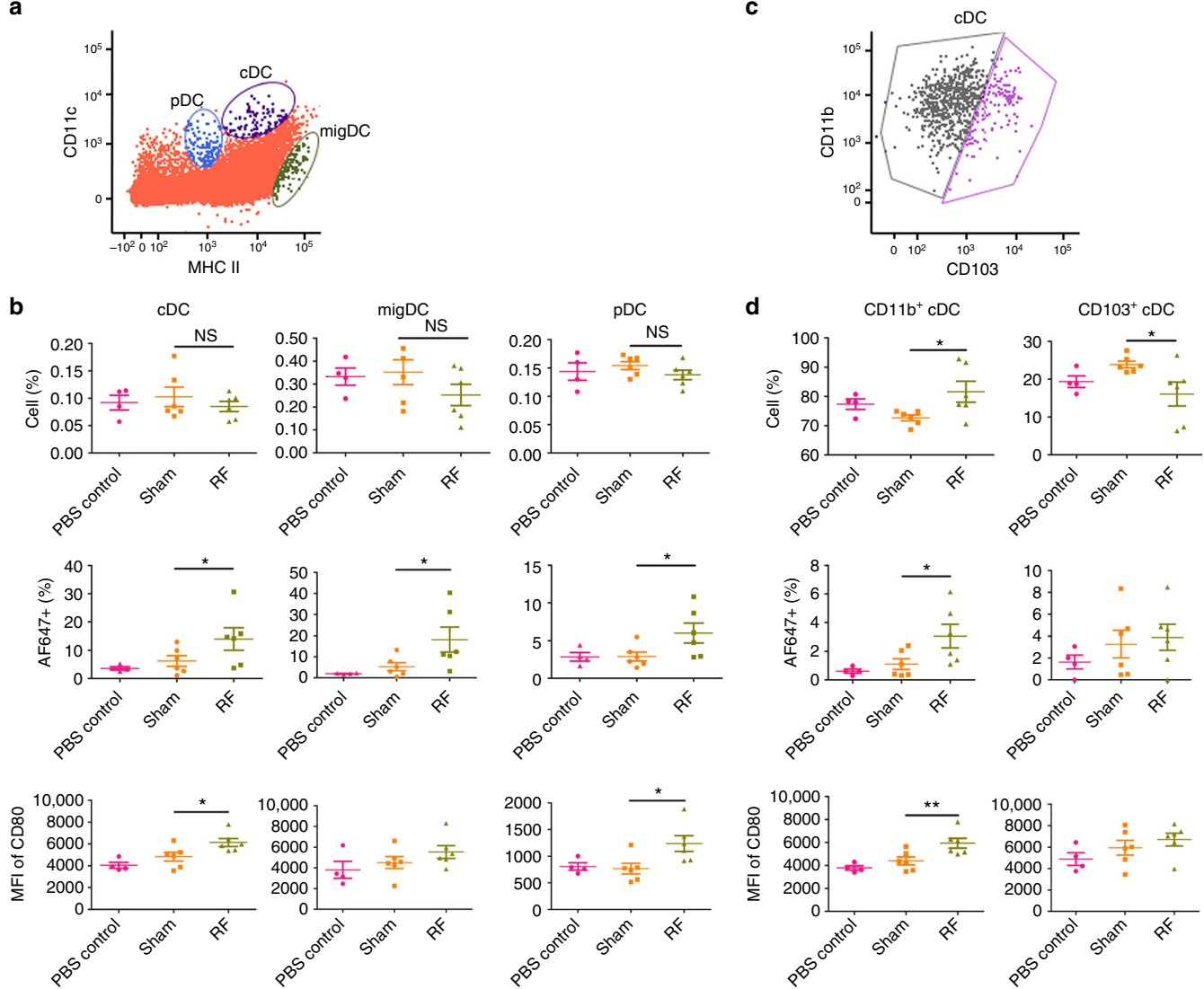

**Fig. 3** RF increases antigen uptake and maturation of DCs in dLNs. DLNs were collected at the same time when skin was collected in Fig. 2 and then passed through cell strainers to prepare single-cell suspensions. Cells were then stained with fluorescence-conjugated antibodies and subjected to flow cytometry analysis. **a** Live cells were first gated and then plotted based on CD11c and MHC II expression. cDC, migDC, and pDC were gated as MHC II^int CD11c^hi, MHC II^hi CD11c^int, and MHC II^low CD11c^low, respectively. **b** Percentage of cDC, migDC, and pDC was shown in upper panels. Percentage of AF647^+ cells in cDC, migDC, and pDC was shown in middle panels. MFI of CD80 in cDC, migDC, and pDC was shown in lower panels. **c** cDC were further gated into CD11b^+ and CD103^+ subsets based on CD11b and CD103 expression. **d** Percentage of CD11b^+ and CD103^+ cDC as well as percentage of AF647^+ cells and MFI of CD80 in CD11b^+ and CD103^+ cDC were shown in upper, middle, and lower panels, respectively. $n = 4$ for PBS control and 6 for sham and RF groups. Student's t-test was used to compare differences between RF and sham groups. *$p < 0.05$; **$p < 0.01$. Representative of two independent experiments

RFA was further compared with CpG, a Th1 adjuvant capable of inducing potent CD8^+ T-cell responses[29]. After prime/boost immunization, we found RFA and CpG significantly increased anti-OVA antibody titer by ~ 15 and 11 folds, respectively (Fig. 5a). Antibody subtype analysis found RFA mainly enhanced IgG1 (Fig. 5b) but not IgG2c antibody titer (Fig. 5c). In comparison, CpG adjuvant mainly enhanced IgG2c (Fig. 5c) but not IgG1 antibody titer (Fig. 5b). These results indicated that RFA and CpG induced Th2- and Th1-biased immune responses, respectively. We further found combination of RFA and CpG (RFA/CpG) could induce more potent anti-OVA antibody titer than either adjuvant alone (Fig. 5a). The ability of RFA to boost anti-OVA antibody production and the synergistic effects between RFA and CpG were also observed in BALB/c mice (Supplementary Fig. 5). Besides humoral immune responses, CD8^+ T-cell responses were evaluated by intracellular cytokine

staining of splenocytes 1 week after boost. As shown in Figs. 5d and 5e, RFA increased OVA-specific IL4- and IFNγ-secreting CD8^+ T cells by 129% and 96%, respectively, whereas CpG increased OVA-specific IL4- and IFNγ-secreting CD8^+ T cells by 81% and 65%, respectively. RFA/CpG more significantly increased IL4-secreting CD8^+ T cells by 423% (Fig. 5d). No significant difference could be found in IFNγ-secreting CD8^+ T cells among RFA, CpG, and RFA/CpG groups (Fig. 5e). A second set of mice were similarly immunized and challenged with E.G7-OVA. As shown in Fig. 5f, tumor growth was significantly inhibited in RFA and CpG groups and more significantly inhibited in RFA/CpG group. All mice succumbed to tumor within 2 weeks in non-immunized and no adjuvant groups (Fig. 5g). 20% and 10% mice were tumor-free in RFA and CpG group, respectively, whereas 50% mice were tumor-free in RFA/CpG group (Fig. 5g). These results indicated that RFA had similar

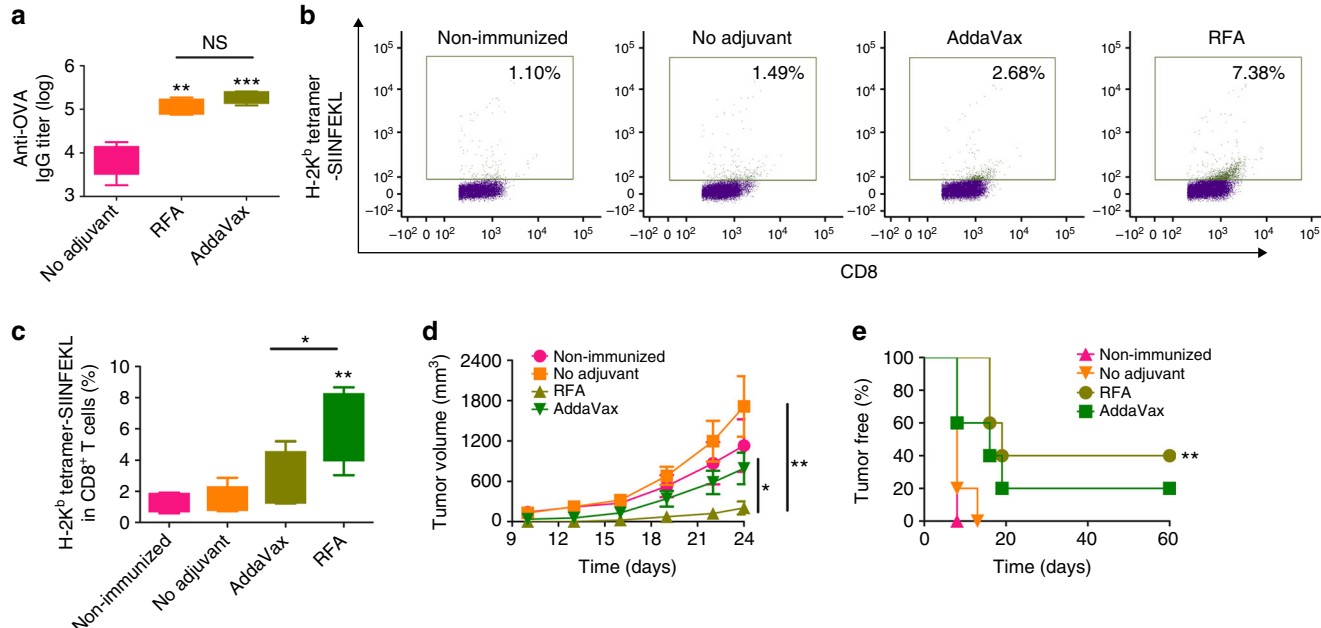

**Fig. 4** Comparison of RFA with AddaVax to boost ID OVA-induced humoral and cellular immune responses. C57BL/6 mice were intradermally immunized with 10 μg OVA alone or in the presence of RFA, or intramuscularly immunized with 10 μg OVA in the presence of AddaVax. Immunization was repeated 2 weeks later. **a** Serum anti-OVA antibody titer was measured 2 weeks after boost. **b**, **c** PBMCs were isolated one week after boost, stimulated with OVA followed by staining with fluorescence-conjugated anti-CD4, anti-CD8 antibodies, and H-2K$^b$-restricted OVA$_{257-264}$ tetramer. Frequency of tetramer$^+$ CD8$^+$ T cells was analyzed by flow cytometry. Cells were first gated based on CD4 and CD8 expression and CD8$^+$CD4$^-$ cells were further analyzed based on tetramer staining. Representative dot plots were shown in **b** and percentage of tetramer$^+$CD8$^+$ T cells were shown in **c**. **d**, **e** Two weeks after boost, mice were challenged with 5 × 10$^5$ E.G7-OVA cells. Tumor growth (**d**) and percentage of tumor-free mice (**e**) were monitored for total 60 days. $n = 5$. One-way ANOVA with Tukey's multiple comparison test was used to compare differences between groups in **a** and **c**. Two-way ANOVA with Bonferroni post test was used to compare tumor growth in **d**. Log-rank test with Bonferroni correction was used to compare differences between adjuvant (RFA or AddaVax) and no adjuvant groups in **e**. *$p < 0.05$; **$p < 0.01$; ***$p < 0.001$. NS not significant. Representative of two independent experiments

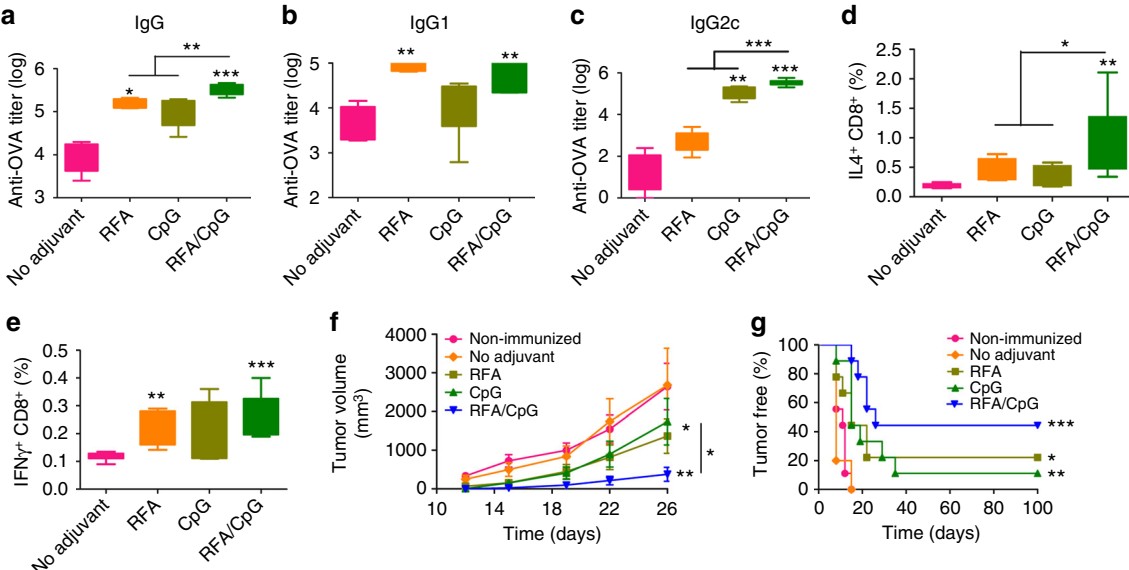

**Fig. 5** Comparison of RFA with CpG to boost ID OVA-induced humoral and cellular immune responses. C57BL/6 mice were intradermally immunized with 10 μg OVA alone or in the presence of RFA, CpG, or RFA/CpG. Immunization was repeated 2 weeks later. **a**–**c** Serum anti-OVA IgG (**a**), and subtype IgG1 (**b**), and IgG2c antibody titer (**c**) were measured 2 weeks after boost. **d**, **e** Splenocytes were prepared one week after boost, stimulated with OVA followed by intracellular cytokine staining and flow cytometry analysis. Percentage of IL4 and IFNγ-secreting CD8$^+$ T cells was shown in **d** and **e**, respectively. **f**, **g** Another set of mice were similarly immunized as above and then subcutaneously challenged with 10$^6$ E.G7-OVA cells 2 weeks after boost. Tumor volume (**f**) and percentage of tumor-free mice (**g**) were monitored for total 100 days. $n = 8$–9. One-way ANOVA with Tukey's multiple comparison test was used to compare differences between groups in **a**–**e**. Two-way ANOVA with Bonferroni post-test was used to compare differences in **f**. Log-rank test with Bonferroni correction was used to compare differences between adjuvant (RFA, CpG, or RFA/CpG) and no adjuvant groups in **g**. *$p < 0.05$; **$p < 0.01$; ***$p < 0.001$. NS not significant. Representative of two independent experiments

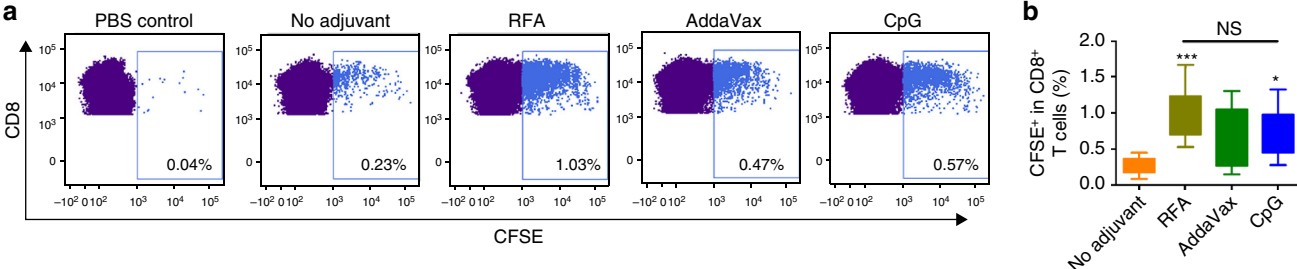

**Fig. 6** RFA stimulates significant OT-I T-cell proliferation in vivo. CFSE-stained OT-I T cells were adoptively transferred to syngeneic C57BL/6 mice followed by ID immunization of 10 μg OVA alone or in the presence of RFA, AddaVax, or CpG adjuvant 24 h later. ID injection of PBS served as control. DLNs were harvested 4 days later and analyzed for proliferation of OT-I T cells by flow cytometry. Live cells were gated and then plotted based on CD4 and CD8 expression. CD8+CD4− T cells were analyzed for CFSE levels. **a** Representative dot plots showing percentage of CFSE+ cells in CD8+ T cells. **b** Percentage of CFSE+ cells in CD8+ T cells of different groups. n = 8–13. One-way ANOVA with Tukey's multiple comparison test was used to compare differences between groups. *p < 0.05; ***p < 0.001. NS, not significant. Representative of three independent experiments

adjuvant potency to CpG in stimulation of OVA-specific CD8+ T-cell production.

The ability of RFA to stimulate OVA-specific CD8+ T-cell responses was further explored by adoptive transfer of carboxy-fluorescein succinimidyl ester (CFSE)-stained OT-I T cells to syngeneic C57BL/6 mice. As shown in Fig. 6a, OVA immunization alone stimulated low levels of OT-I T-cell proliferation, whereas incorporation of RFA induced vigorous OT-I T-cell expansion. The ability of RFA to stimulate OT-I T-cell proliferation was at least comparable to CpG and AddaVax adjuvants (Fig. 6b). The above data indicated that RFA possessed potent humoral and cellular adjuvant effects to boost OVA immunization.

**RFA augments rHA-induced humoral and cellular immunity.** rHA of influenza A/California/07/2009 (H1N1) was then used as a different protein antigen to explore RFA effects. RFA was further compared with AddaVax and Alum to boost rHA immunization considering the same types of adjuvants have been approved to boost seasonal and/or pre-pandemic influenza vaccination[6,30]. After prime/boost immunization, we found rHA-specific IgG antibody titer was significantly increased in RFA group as compared with that in no adjuvant group (Fig. 7a). rHA-specific IgG antibody titer in RFA group showed no significant difference when compared with that in AddaVax group, but was significantly higher than that in Alum group (Fig. 7a). rHA-specific IgG1 antibody titer showed the same trend to total IgG antibody titer (Fig. 7b). rHA-specific IgG2c antibody titer was significantly higher in RFA group than that in AddaVax or Alum group (Fig. 7c). More significant induction of IgG1 as compared to IgG2c antibody titer indicated that RFA induced Th2-biased immune responses (Fig. 7b, c). rHA-specific CD8+ T-cell responses were then analyzed by intracellular cytokine staining of PBMCs 1 week after boost. As shown in Figs. 7d and 7e, RFA significantly increased rHA-specific IFNγ-secreting CD8+ T cells by ~ 160%, whereas AddaVax and Alum failed to increase IFNγ-secreting CD8+ T cells. These results indicated that RFA was comparable to AddaVax to enhance rHA-specific antibody responses and more potent than AddaVax to enhance rHA-specific CD8+ T-cell responses. Our data also indicated RFA was more potent than Alum to enhance rHA-specific humoral and cellular immune responses.

**RFA boosts pdm09 vaccination.** Pdm09 vaccine composed of highly purified surface antigens was used to further evaluate RFA effects. Mice were intradermally immunized with 0.3 or 0.06 μg pdm09 vaccine in the presence or absence of RFA or intramuscularly immunized with the same vaccine dose in the presence of

AddaVax. Serum hemagglutination inhibition (HAI) titer was measured 3 weeks later. Serum HAI titer was significantly increased from 9 in no adjuvant group to 31 in RFA group and 35 in AddaVax group (Fig. 8a). No significant difference in HAI titer could be found between RFA and AddaVax groups (Fig. 8a). ID immunization in the presence of RFA induced minimal local reactions as evidenced by lack of skin damage or color change (Supplementary Fig. 6a). Rectal temperature and serum cytokine levels were measured as indicators of systemic reactions. No significant difference in rectal temperature was found in all groups before immunization (left panel, Supplementary Fig. 6b). Rectal temperature slightly but significantly increased in RFA and Adda-Vax groups 6 h after immunization despite the lack of a significant difference among no adjuvant, RFA, and AddaVax groups (middle panel, Supplementary Fig. 6b). Rectal temperature then returned to baseline levels in RFA and AddaVax groups at 24 h (right panel, Supplementary Fig. 6b). Serum tumor necrosis factor α (TNFα) remained at baseline levels in all groups (left panel, Supplementary Fig. 6c). Serum IL6 slightly increased in all immunization groups and yet showed no significant difference when compared to that in non-immunized group (right panel, Supplementary Fig. 6c). These results indicated that RFA had a good local and systemic safety to boost pdm09 vaccination. After lethal viral challenges, significant protection was only induced in RFA and AddaVax groups (Fig.8b, c). All mice died in 6 days in non-immunized group and in 10 days in no adjuvant group, whereas ~ 60% mice survived in RFA and AddaVax groups (Fig. 8b). Body weight of live mice gradually recovered to their original levels in RFA and AddaVax groups (Fig. 8c). Significant HAI titer (~ 23) was also induced in RFA group when vaccine dose was reduced to 0.06 μg, whereas AddaVax completely lost its adjuvanticity at this vaccine dose (Fig. 8d). Following lethal viral challenges, significant protection was only observed in RFA group (Fig. 8e). All mice died in 9 days in non-immunized, no adjuvant, and AddaVax adjuvant groups, while 60% mice in RFA group survived (Fig.8e). Body weight of live mice in RFA group gradually recovered to less than 5% below their original levels (Fig. 8f). Our studies indicated that RFA could significantly boost pdm09 vaccination with adjuvant effects comparable or better than AddaVax adjuvant depending on vaccine doses.

**RFA stimulates HSP70 synthesis and activates MyD88 pathway.** RFA likely induces tissue stress and release of endogenous danger signals to mediate its adjuvant effects. We first explored whether thermal heating was crucial for RFA effects. As shown in Supplementary Fig. 7, pre-cooling of local skin and RF tips significantly reduced RFA effects. With prior cooling, RFA failed to significantly increase anti-OVA antibody titer after prime

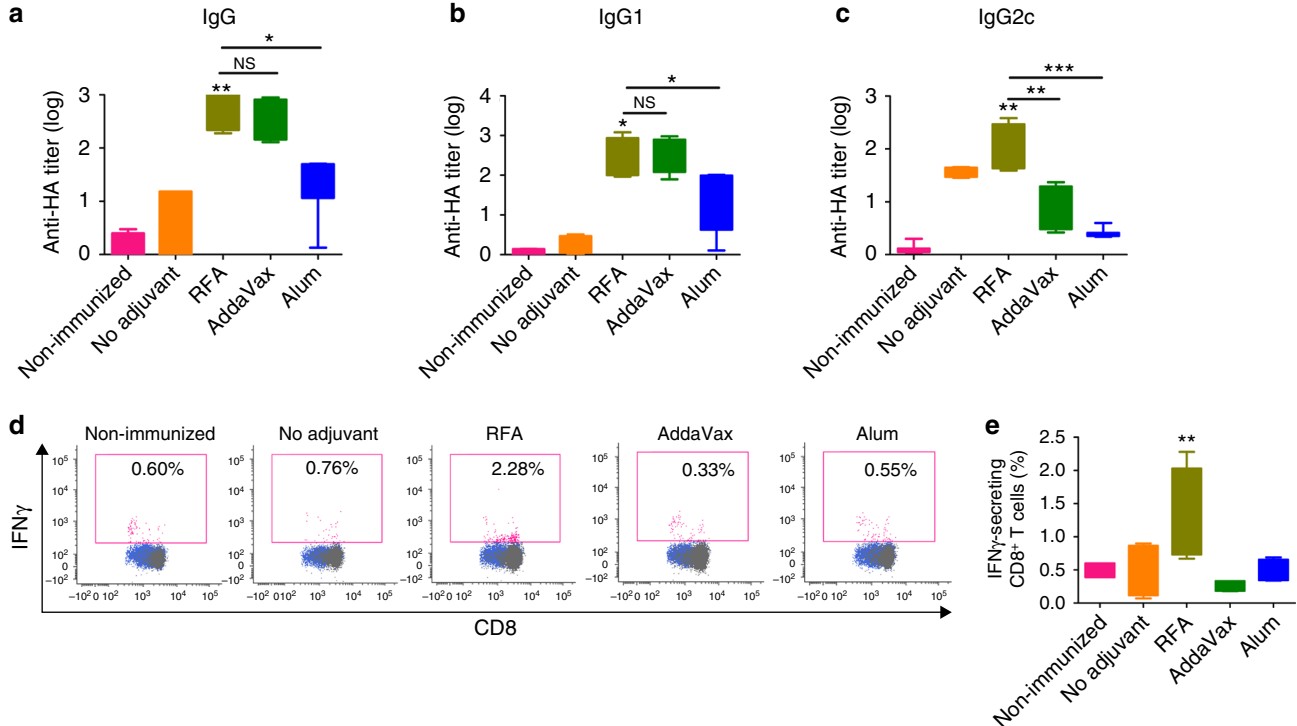

**Fig. 7** RFA increases rHA-induced humoral and cellular immune responses. C57BL/6 mice were intradermally immunized with 5 μg rHA alone or in the presence of RFA, or intramuscularly immunized with 5 μg rHA in the presence of AddaVax or Alum. Immunization was repeated 2 weeks later. **a–c** Serum rHA-specific IgG (**a**), and subtype IgG1 (**b**) and IgG2c antibody titer (**c**) measured 2 weeks after boost. **d, e** PBMCs were isolated 1 week after boost, stimulated with rHA followed by intracellular cytokine staining and flow cytometry analysis of percentage of IFNγ-secreting CD8+ T cells. Representative dot plots were shown in **d** and percentage of IFNγ-secreting CD8+ T cells was shown in **e**. n = 4–6. One-way ANOVA with Tukey's multiple comparison test was used to compare differences between groups. *$p < 0.05$; **$p < 0.01$; ***$p < 0.001$. NS not significant. Representative of two independent experiments

(Supplementary Fig. 7a) and only increased anti-OVA antibody titer by fivefolds after boost (Supplementary Fig. 7b). Thermal heating may induce release of heat shock proteins (HSPs) to mediate vaccine-induced immune responses. HSPs are highly abundant intracellular proteins with important chaperone functions and can be classified into about ten families based on molecular weight and intracellular locations[31]. HSP70 is among the mostly explored HSPs with potent vaccine adjuvant effects[31]. As HSP70 has two isoforms (inducible HSP70 and constitutive HSc70)[32], we explored impacts of RFA on protein levels of both isoforms. As shown in Fig. 9a, RFA significantly increased HSP70 but not HSc70 levels at 6 and 24 h. To explore whether RFA effects were specific to HSP70, we evaluated protein levels of inducible HSP90[33]. We found RFA failed to increase HSP90 levels (Fig. 9a). Increased HSP70 levels were further confirmed by immunohistochemistry (IHC) analysis. In non-treated skin, positive HSP70 staining was found in epidermis and specific dermal structures, like hair follicles and sebaceous glands (Fig. 9b). In contrast, more uniform and intense HSP70 staining was found across entire epidermis and dermis of RFA-treated skin (Fig. 9b). Considering HSP70 can bind to Toll-like receptor (TLR) 2 and TLR4 and MyD88 is located downstream of these TLRs[34,35], we explored whether MyD88 was crucial for RFA effects. As shown in Fig. 9c, lack of MyD88 significantly reduced RFA effects. RFA only increased anti-OVA antibody titer by ~ 2.5 folds in MyD88 knockout (KO) mice, whereas RFA increased anti-OVA antibody titer by over 25 folds in wild-type (WT) mice (Fig. 9c). In further support, peptide inhibitor of MyD88 but not control peptide significantly inhibited RFA effects (Fig. 9d). Serum anti-OVA antibody titer was reduced by more than 70%

by MyD88-specific peptide and only ~ 10% by control peptide (Fig. 9d). NALP3 inflammasome represents another signaling pathway that can be activated by a variety of endogenous danger signals[36–38]. We found lack of NALP3 had no significant impacts on RFA effects (Supplementary Fig. 8). Our data indicated RFA stimulated HSP70 synthesis and activated MyD88 to mediate its adjuvant effects.

## Discussion
This study, for the first time, proved that non-invasive RF treatment could significantly boost protein/subunit vaccine-induced humoral and cellular immune responses with RFA effects non-inferior to widely used chemical adjuvants. RFA significantly increased OVA-induced antibody production with adjuvant effects comparable to AddaVax (Fig. 4a) and CpG adjuvant (Fig. 5a), and enhanced rHA-induced antibody production with adjuvant effects comparable to AddaVax and superior to Alum adjuvant (Fig. 7a). RFA also significantly increased pdm09 vaccine-induced HAI titer with adjuvant effects comparable to AddaVax at 0.3 μg vaccine dose (Fig. 8a). RFA also significantly increased pdm09 vaccine-induced HAI titer when vaccine dose was reduced to 0.06 μg (Fig. 8d). Interestingly, AddaVax completely lost its adjuvanticity at this low vaccine dose (Fig. 8d). Consistent with HAI titer, significant protection against lethal viral challenges was observed in RFA and AddaVax groups at 0.3 μg vaccine dose (Fig. 8b, c) and only in RFA group at 0.06 μg vaccine dose (Fig. 8e, f). Remarkably, a similar survival rate was observed in RFA group when vaccine dose was reduced from 0.3 to 0.06 μg (Figs. 8b and 8e), hinting RFA may possess

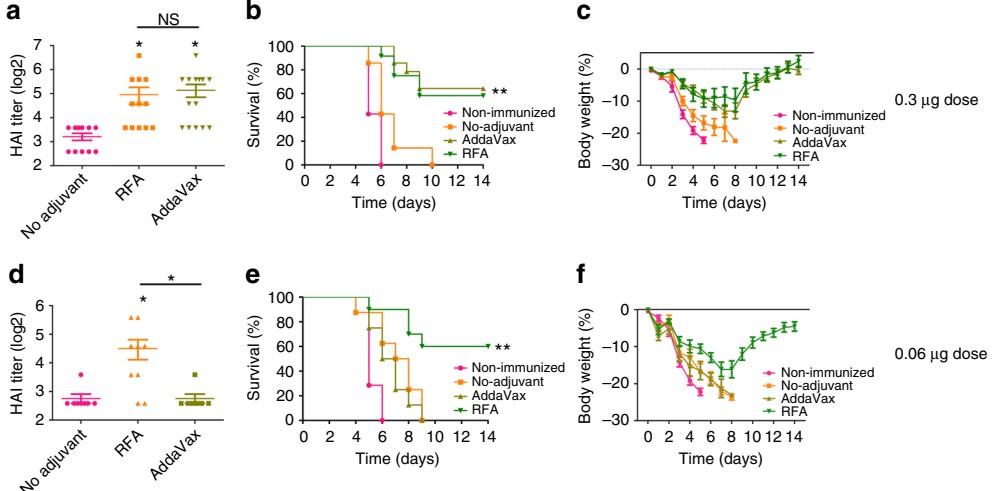

**Fig. 8** RFA boosts pdm09 vaccination. **a–c** C57BL/6 mice were intradermally immunized with 0.3 μg pdm09 vaccine alone (no adjuvant) or in the presence of RFA (RFA), or intramuscularly immunized with the same vaccine dose in the presence of AddaVax (AddaVax), or intradermally injected with the same volume of PBS (non-immunized). Serum HAI titer was measured 3 weeks later (**a**). Mice were then intranasally challenged with $10 \times LD_{50}$ of mouse-adapted pdm09 viruses. Survival and body weight loss were monitored daily for 14 days and shown in **b** and **c**, respectively. **d–f** C57BL/6 mice were similarly immunized as above at 0.06 μg vaccine dose. HAI titer was measured 3 weeks later (**d**). Mice were then challenged with $10 \times LD_{50}$ of mouse-adapted pdm09 viruses. Survival and body weight loss were similarly monitored and shown in **e** and **f**, respectively. $n = 12$–14 in **a**–**c** and 8–9 in **d**–**f**. One-way ANOVA with Tukey's multiple comparison test was used to compare differences between groups in **a** and **d**. Log-rank test with Bonferroni correction was used to compare differences between adjuvant (RFA, AddaVax) and no adjuvant groups in **b** and **e**. *$p < 0.05$; **$p < 0.01$. NS not significant. Representative of two independent experiments

significant dose-sparing effects to boost pdm09 vaccination. Despite of similar survival rates, high vaccine dose (0.3 μg) in the presence of RFA has been associated with less maximum body weight loss (10% vs. 16%) and more complete body weight recovery (+ 2.4% vs. − 4.5%) as compared with low vaccine dose (0.06 μg) in the presence of RFA (Fig. 8c–f). These results indicated that the physical RFA was at least comparable to the highly potent MF59-like AddaVax adjuvant to boost pdm09 vaccination.

Besides humoral adjuvant effects, RFA also possessed potent cellular adjuvant effects as evidenced by strong induction of OVA and rHA-specific CD8+ T-cell responses (Figs. 4c, 5d, 5e, 7e). The ability of RFA to stimulate OVA-specific CD8+ T cells was also confirmed by adoptive transfer of CFSE-labelled OT-I T cells to syngeneic mice followed by OVA immunization in the presence of RFA. We found vigorous expansion of CFSE-labelled OT-I T cells in RFA group but not in no adjuvant group (Fig. 6). RFA also showed at least similar potency to AddaVax and CpG adjuvants to stimulate proliferation of OVA-specific OT-I T cells (Fig. 6). Consistent with strong induction of OVA-specific CD8+ T-cell responses, OVA immunization in the presence of RFA but not OVA immunization alone significantly inhibited E.G7-OVA tumor growth (Figs. 4d, 5f). Protein antigens generally don't stimulate CD8+ T-cell responses considering they are mainly presented on MHC class II molecules and stimulate CD4+ T-cell responses[39]. It's intriguing how protein antigens gain access to stimulate CD8+ T-cell responses following RFA treatment. One explanation is that HSP70 induced after RFA treatment facilitates cross-presentation of protein antigens on MHC class I molecules for stimulation of CD8+ T-cell responses[31]. The ability of RFA to simultaneously boost both arms of adaptive immunity is promising to improve overall protective efficacy of vaccines due to the differential roles of humoral and cellular immunity in prevention and elimination of virus-infected cells. In pdm09 vaccine challenge studies, we observed similar rates of body weight loss (~ 9%) in the first 5 days in RFA and AddaVax

groups (Fig. 8c), in line with similar HAI titers induced in both groups (Fig. 8a). After 5 days, no further body weight loss was observed in RFA group, while more body weight loss was observed in AddaVax group (Fig. 8c). This led to less maximum body weight loss in RFA group as compared with that in AddaVax group (9% vs. 13%). Less maximum body weight loss in RFA group may be due to the induction of vaccine-specific CD8+ T-cell responses to eliminate virus-infected cells, which remains to be explored in the near future.

RFA also has a good safety to boost ID vaccination. RFA induced transient low-level local inflammation, whereas chemical adjuvants induced persistent strong local inflammation (Fig. 1). Pdm09 vaccination in the presence of RFA induced minimal local and systemic reactions (Supplementary Fig. 6). Despite a slight increase of rectal temperature at 6 h in RFA and AddaVax groups, rectal temperature returned to baseline levels at 24 h in both groups (Supplementary Fig. 6). Besides the good local and systemic safety, the physical RFA introduces no foreign materials into the body and is less likely to induce Auto-immune/Inflammatory Syndromes Induced by Adjuvants, a term recently coined to describe rare clinical conditions (i.e., gulf-war syndromes, macrophagic myofasciitis, and narcolepsy) associated with immunization of chemical adjuvant-containing vaccines[40].

Regarding cellular adjuvantation mechanisms, we found RFA significantly improved antigen uptake and maturation of DCs in skin and dLNs (Figs. 2, 3, Supplementary Fig. 3, 4). In skin, we found dermal Langerin+CD11b−CD103+ and Langerin−CD11b− DCs improved antigen uptake following RFA treatment (Fig. 2c). In dLNs, CD11b+ but not CD103+ cDC improved antigen uptake and maturation (Fig. 3d) and all migDC subsets improved antigen uptake but not maturation following RFA treatment (Supplementary Fig. 4b). Although laser adjuvant also improved DC function[16–18], DC subsets that responded to laser treatment differed from that responded to RF treatment. In one recent study, near-infrared laser was found to modulate migDC but not cDC or

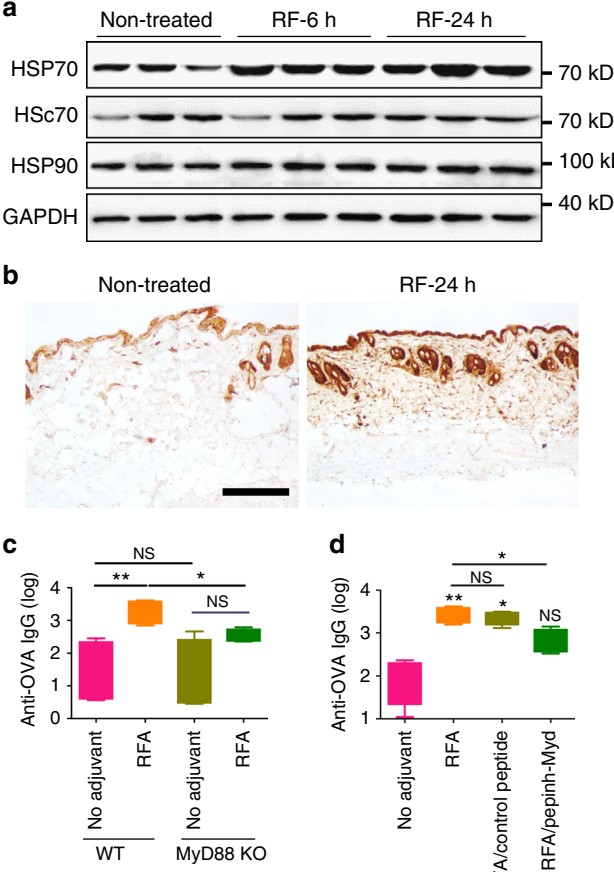

**Fig. 9** RFA increases HSP70 levels and activates MyD88. **a** C57BL/6 mice were exposed to RF and skin HSP70, HSc70, and HSP90 levels were then analyzed by western blotting at 6 and 24 h (hr) using GAPDH as internal control. **b** Representative IHC images of HSP70 expression in RF-treated and non-treated skin at 24 h. Scale: 250 μm. **c** WT and MyD88 KO mice were exposed to RF or sham treatment followed by ID injection of 10 μg OVA into RF or sham-treated skin. Serum anti-OVA antibody titer was measured 2 weeks later. $n = 4$–5. **d** WT mice were intradermally injected with 100 μg Pepinh-Control or Pepinh-MyD, or the same volume of PBS 3 and 1 h before RF treatment and ID OVA immunization at 10 μg dose. OVA immunization alone served as control (No adjuvant). Serum anti-OVA antibody titer was measured 2 weeks later. $n = 4$–5. One-way ANOVA with Tukey's multiple comparison test was used to compare differences between groups in **c** and **d**. $*p < 0.05$; $**p < 0.01$. NS not significant. Representative of two independent experiments in **c** and **d**

pDC to mediate its adjuvant effects[18]. In addition, laser adjuvant was largely non-inflammatory without induction of significant local inflammation[16–18]. These divergences may explain the different adjuvant effects induced by laser and RF[16–18].

Regarding tissue stress, we found thermal stress played a crucial roles in RFA effects as we found pre-cooling of local skin and RF tips could significantly reduce RFA effects especially in primary immunization (Supplementary Fig. 7). Regarding endogenous danger signals, we found RFA significantly increased HSP70 levels (Fig. 9a, b). HSP70 is able to bind hydrophobic regions of antigenic peptides, increase antigen presentation on MHC class I and II molecules, and enhance CD4+ and CD8+ T-cell responses[31]. In addition, only a small amount of peptides (a few hundred pictograms) are required to induce potent immune responses when complexed with HSP70[31]. The ability of RFA to boost both humoral and cellular immune responses (Figs. 4, 5, 7) and elicit

significant HAI titer at low pdm09 vaccine dose (0.06 μg) (Fig. 8d) is in line with the above features of HSP70. Furthermore, we found MyD88 played a crucial role in RFA effects (Fig. 9c, d), which is also in line with the literature that HSP70 can bind to TLR2 and TLR4 and MyD88 is located downstream of these TLRs[31,41]. Our study is also consistent with a recent study, in which a microneedle patch loaded with tumor lysate and melanin was exposed to near-infrared light to generate transdermal heat to promote anti-tumor immune responses[42]. In that study, heat generation was associated with increase of HSP70 expression[42]. That study and our work support induction of dermal heating and HSP70 release to boost vaccine-induced immune responses.

With increasing use of protein/subunit vaccines, novel adjuvants are highly demanded to improve vaccine efficacy and aid in development of new vaccines. Considering protein/subunit vaccines often lack the ability to induce CD8+ T-cell immune responses, ideal adjuvants better augment both humoral and cellular immune responses to improve overall protective efficacy of vaccines. RFA is promising to be one such adjuvant with good safety and potency to boost both arms of adaptive immunity. RF also has unique advantages to serve as vaccine adjuvant considering RF energy can penetrate deep into the tissue (at least a few millimeters) and RF energy absorption is not affected by skin color[43]. RFA can conveniently boost vaccination without pre-mixing with vaccines and thus requires no modification of vaccine manufacturing.

## Methods
**Reagents**. OVA (A5503) and MPL (L6895) were purchased from Sigma (St. Louis, MO). AddaVax, Pepinh-MYD, and Pepinh-Control peptides were purchased from Invivogen (San Diego, CA). CpG 1826 was synthesized by Trilink Biotechnologies (San Diego, CA). Alum (Imject) was ordered from Thermo Fisher Scientific (Rockford, lL). rHA of Influenza A/California/07/2009 (H1N1) (FR559) was obtained from International Reagent Resource (IRR, Manassas, VA). Monovalent 2009 H1N1 influenza (pdm09) vaccine (NR-20083) was obtained from BEI Resources (Manassas, VA). Fluorescence-conjugated antibodies used in immunostaining and flow cytometry were purchased from eBiosciences, Biolegends, or Affymetrix, all from San Diego, CA.

**Mice**. C57BL/6 and BALB/c mice (6–8 weeks old, male) were purchased from Charles River Laboratories (Wilmington, MA). MyD88 and NALP3 KO mice and OT-I transgenic mice were obtained from Jackson Laboratory (Bar Harbor, ME). Animals were housed in animal facilities of University of Rhode Island (URI) and anesthetized for hair removal, RF treatment, and immunization. Animal experiments involving influenza viruses were conducted in animal biosafety level 2 (ABSL2) facility of URI. All animal procedures were approved by the Institutional Animal Care and Use Committee of URI.

**RF device**. A cosmetic fractional bipolar RF device of ~ 1 MHz (Norlanya Technology Co., Hong Kong, China) equipped with 12 × 12 array of microelectrodes in 2 × 2 cm² area was used. This device has three energy settings (low, medium, high) and high-energy setting was used in this study to induce significant tissue stress in 1–2 min. For RF treatment, a thin layer of ultrasound coupling medium was applied on the skin surface as recommended by manufacturer and RF device was then firmly pressed to allow treatment tips to have a close contact with skin surface.

**Immunization**. Hair on the lateral dorsal skin of mice was shaved and completely removed with the help of a hair removal lotion (Nair). Next day, hair-removed skin was exposed to RF or sham treatment for indicated time (RF was not activated in sham treatment) followed by ID injection of 10 μg OVA, 5 μg rHA, 0.3 or 0.06 μg pdm09 vaccine into RF- or sham-treated skin. Endotoxin levels of OVA was reduced with Detoxi-Gel Endotoxin Removing Column (20344, Thermo Fisher Scientific) from original 60 EU/ml to below 15 EU/ml for in vivo application[44]. Endotoxin levels were measured by Pierce LAL Chromogenic Endotoxin Quantification Kit (88282, Thermo Fisher Scientific). In chemical adjuvant groups, mice were ID injected with the same amount of antigen in the presence of 30 μg CpG, or intramuscularly injected with the same amount of antigen in the presence of AddaVax (50%, vol/vol) or Alum (1:1 volume ratio) except otherwise specified. Intramuscular delivery of AddaVax and Alum in most of the experiments was to avoid significant local reactions following ID delivery of the two adjuvants as revealed in our pilot

studies. For combinatorial adjuvant groups, mice were exposed to RF or sham treatment followed by ID injection of 10 μg OVA in the presence of 30 μg CpG.

**Antibody titer measurement**. Serum antibody titer was measured by enzyme-linked immunosorbent assay (ELISA). In brief, vaccine antigens (1–10 μg/ml) were coated into 96-well ELISA plates at 4 °C overnight. After blocking with 5% non-fat milk, 2-serial dilutions of immune sera were added and incubated at room temperature for 90 min. After washing in phosphate-buffered saline (PBS) supplemented with 0.05% Tween 20 (PBST), horse-radish peroxidase (HRP)-conjugated sheep anti-mouse IgG secondary antibodies (1:5000, NA931, GE Healthcare Life Sciences) were added and incubated at room temperature for 1 h. After washing in PBST, TMB substrates were added and reactions were then stopped by addition of 3 M $H_2SO_4$. Optical absorbance (OD450/490 nm) was read in a microplate reader (Molecular Device). Serum antibody titer was defined as the reciprocal dilution factor that resulted in OD450/490 nm that was ∼ 3 times higher than the background values. For detection of subtype antibody titer, HRP-conjugated anti-mouse IgG1 and IgG2a or IgG2c secondary antibodies are used.

**Cell-mediated immune response**. To measure vaccine-specific $CD4^+$ and $CD8^+$ T cells in PBMCs, small amount of blood (∼ 50 μl) was collected into heparinized tubes followed by centrifugation at 1,300 rpm for 5 min. After red blood cell (RBC) lysis, PBMCs then were stimulated with 10 μg/ml OVA or rHA in the presence of 4 μg/ml anti-CD28 antibodies (37.51) overnight. Cells were harvested and stained with fluorescence-conjugated H-2K$^b$-restricted OVA$_{257-264}$ (SIINFEKL) tetramer (NIH tetramer core facility), anti-CD4 (GK1.5), and anti-CD8 antibodies (53–6.7). For intracellular cytokine staining, Brefeldin A (420601, Biolegend) was added 5 h before cell harvest. PBMCs were then stained with fluorescence-conjugated anti-CD4 and anti-CD8 antibodies, fixed and permeabilized, and then stained with fluorescence-conjugated anti-IFNγ (XMG1.2) and anti-IL4 antibodies (11B11). To detect OVA-specific $CD4^+$ and $CD8^+$ T cells in splenocytes, single-cell suspensions were prepared by passing spleen through 40 μm cell strainers. After RBC lysis, splenocytes were stimulated with 10 μg/ml OVA in the presence of anti-CD28 antibodies overnight. Brefeldin A was added 5 h before cell harvest. Splenocytes were similarly stained with fluorescence-conjugated anti-CD4, CD8, IFNγ, and IL4 antibodies. Cells were then subjected to flow cytometry analysis in BD FACSVerse.

**Tumor model**. E.G7-OVA cells (CRL-2113, ATCC) were cultured in PRMI 1640 media supplemented with 10% fetal bovine serum, 2 mM ʟ-glutamine, and 100 U/ml penicillin and 100 μg/ml streptomycin. Cells were harvested at ∼ 80% confluency, washed in PBS, and $5 \times 10^5$ or $1 \times 10^6$ cells were subcutaneously injected into right flank of C57BL/6 mice. Tumor size was measured with a digital caliper and tumor volume was calculated with the following formula: $v = \frac{1}{2}ab^2$, where $a$ and $b$ are long and short diameter of the tumor, respectively.

**HAI titer**. Serum HAI titer was measured following a well-established protocol[45]. In brief, serum samples were incubated with receptor-destroying enzyme II and then heat inactivated to remove complement activities. Serum samples were further adsorbed with chicken RBCs to remove non-specific binding. Serum samples were subjected to a twofold serial dilution and incubated with four hemagglutinating units of pdm09 viruses (A/California/07/2009). Pdm09 viruses were obtained from IRR (FR-201) and propagated in 9–11-day embryonic eggs for use in this study. Chicken RBCs were added and HAI titer was determined as the reciprocal of the highest dilution that completely inhibited agglutination of chicken RBCs.

**Mouse adaptation of pdm09 viruses**. Pdm09 viruses were adapted in C57BL/6 mice for use in challenge studies[45]. In brief, mice were infected by intranasal instillation of 20 μl influenza viruses at ∼ $10^5$ 50% tissue culture infective dose (TCID$_{50}$). Lung was aseptically harvested 3 days later and lung homogenates were used as inoculum for the next passage. Adaptation was repeated four times and viral titer of the last lung homogenates was measured by TCID$_{50}$ assay[45]. In brief, influenza viruses were subjected to a 10-fold serial dilution and then incubated with Madin–Darby Canine Kidney cells (CCL34, ATCC) for 4 days. Chicken RBCs were added and agglutination pattern was recorded. TCID$_{50}$ was calculated based on the formula provided in the protocol[45].

**Lethal viral challenge**. LD$_{50}$ of mouse-adapted viruses was first determined[45]. In brief, six groups of mice ($n = 4$) were infected with $10^0$, $10^1$, $10^2$, $10^3$, $10^4$, and $10^5$ TCID$_{50}$ influenza viruses. Survival and body weight were monitored daily for 14 days. LD$_{50}$ was calculated by the method of Reed and Muench[45]. For lethal viral challenge, mice were intranasally inoculated with $10 \times$ LD$_{50}$ of influenza viruses under light anesthesia. Body weight and survival were monitored daily for 14 days. Mice with body weight loss more than 25% were euthanized and regarded as dead.

**Antigen uptake and maturation**. Mice were exposed to RF or sham treatment followed by ID injection of 2 μg AF647-OVA into RF or sham-treated skin. Skin was dissected 18 h later and single-cell suspensions were prepared by digestion in collagenase D and dispase followed by staining with fluorescence-conjugated anti-CD11c (N418), CD40 (1C10), CD80 (16–10A1), and CD86 (GL-1). Cells were subjected to flow cytometry analysis of percentage of AF647$^+$CD11c$^+$ cells and also MFI of CD40, CD80, and CD86 within AF647$^+$CD11c$^+$ cells. For DC subset analysis in skin and dLNs, skin and LN cells were stained with fluorescence-conjugated anti-CD11c (N418), MHC II (M5/114.15.2), Langerin (4C7), CD11b (M1/70), CD103 (2E7), CD8α (53–6.7), and CD80 (16–10A1). Cells were then subjected to flow cytometry analysis of percentage of AF647$^+$ cells as well as MFI of CD80 in different DC subsets in skin and dLNs as reported[18,27,28].

**Adoptive transfer of OT-I T cells**. LNs and spleens were harvested from OT-I transgenic mice (003831, Jackson Laboratories), in which $CD8^+$ T cells recognize OVA residues 257–264 in the context of H-2K$^b$. LNs and spleens were passed through 40 μm cell strainers to prepare single-cell suspensions. After RBC lysis, cells were subjected to magnetic beads-based negative purification of naïve $CD8^+$ T cells with a commercial kit (130–096–543, Miltenyi Biotech). $CD8^+$ T cells from OT-I mice (designated as OT-I cells) were then stained with 5 μM CFSE (C34554, Thermo Fisher Scientific) at 37 °C for 20 min. OT-I cells were thoroughly washed in PBS and then adjusted to $10^7$ cells/ml in PBS. OT-I cells ($10^6$) were then intravenously injected into C57BL/6 mice via lateral tail vein. OVA immunization was conducted 24 h later.

**Western blotting**. Skin was homogenized in T-PER™ Tissue Protein Extraction Reagent (78510, Thermo Fisher Scientific). Total protein levels were quantified by BCA protein assay kit (23227, Thermo Fisher Scientific). Samples that contained the same amount of proteins were subjected to sodium dodecyl sulfate-polyacrylamide gel electrophoresis (SDS-PAGE) separation. Proteins were then transferred to a polyvinylidene difluoride (PVDF) membrane and blocked with 5% non-fat milk at 4 °C overnight. PVDF membrane was then incubated with rabbit anti-HSP70 antibodies (1:2000, AF1663, R&D Systems) (no cross-reactivity with HSc70) at room temperature for 90 min. After washing in Tris-buffered saline (TBS) containing 0.05% Tween 20 (TBST), PVDF membrane was incubated with HRP-conjugated anti-rabbit secondary antibodies (1:2000, 7074P2, Cell Signaling Technology) at room temperature for 1 h. After washing in TBST, PVDF membrane was incubated with Pierce ECL Western Blotting Substrate (32109, Thermo Fisher Scientific). PVDF membrane was imaged under myECL Imager (Thermo Fisher Scientific). PVDF membrane was stripped in stripping buffer (62 mM Tris-HCl, pH 6.8, 2% SDS, 100 mM β-mercaptoethanol) for immune blotting detection of other proteins. In brief, stripped PVDF membrane was blocked, incubated with anti-GAPDH antibodies (1:2000, 5174 s, Cell Signaling Technology), rabbit anti-HSc70 polyclonal antibodies (1:500, AB1427, Abcam) (no cross-reactivity with HSP70), or rabbit anti-HSP90 polyclonal antibodies (1:1000, PA3013, Thermo Fisher Scientific) followed by the same procedures for detection of GAPDH, HSc70, and inducible HSP90 expression. Complete stripping was verified by negative signals obtained from incubation of stripped PVDF membrane with substrates alone or sequential incubation with secondary antibodies and substrates.

**Histology and IHC analysis**. Mice were exposed to RF or sham treatment. Skin was dissected at indicated times, fixed in formalin, and then subjected to paraffin sectioning. Paraffin sections were subjected to standard hematoxylin and eosin staining to visualize microscopic structures or Trichrome staining to visualize dermal collagen levels. For IHC, paraffin sections were deparaffinized and then subjected to heat-induced epitope retrieval with antigen unmasking solution (H-3300, Vector Laboratories). Endogenous peroxidase was depleted with 0.3% hydrogen peroxide. Tissue sections were then incubated with SuperBlock (Thermo Fisher Scientific) and then rabbit anti-mouse polyclonal HSP70 antibodies (1:100, AF1663, R&D Systems) at 4 °C overnight. Tissue sections were then rinsed in TBS for three times followed by incubation with HRP-conjugated anti-rabbit secondary antibodies (K4063, DAKO). After washing in TBS, DAB substrate was added and reaction was stopped 2 min later. Sections were counterstained with 20% Gill III hematoxylin, dehydrated, and then cover slipped. Images were taken under Nikon Eclipse E600 microscope by an investigator unknown of sample groups.

**Real-time PCR analysis**. Adjuvant-treated skin was dissected at indicated times. Total RNA was isolated with Trizol method and reverse transcribed. Relative gene expression of cytokines and chemokines was analyzed by real-time PCR in Applied Biosystems ViiA 7 using *GAPDH* as internal control. PCR primers of each gene refer to Supplementary Table 1.

**Immune cell recruitment**. Skin was digested in 0.2% collagenase D (Roche Diagnostics, Mannheim, Germany) and 0.6U/ml dispase (Gibco, Invitrogen, Carlsbad, CA) in PBS at 37 °C for 3 h with intermittent vortexing. Single-cell suspensions were then passed through 40 μm cell strainer and stained with fluorescence-conjugated antibodies against Ly6C (HK1.4), CD11b (M1/70), CD11c

(N418), MHC class II (m5/114.15.2), Ly6G (1A8), F4/80 (BM8) to identify neutrophils (CD11b$^+$, Ly6G$^{hi}$, Ly6C$^+$, F4/80$^-$), eosinophils (CD11b$^+$, Ly6G$^{int}$, Ly6C$^-$, F4/80$^{int}$), monocytes (CD11b$^+$, Ly6G$^-$, Ly6C$^+$, F4/80$^{int}$), macrophages (CD11b$^+$, F4/80$^{hi}$), and mDCs (CD11b$^+$, CD11c$^+$, MHC II$^+$) as reported[46]. The gating strategy was shown in Supplementary Figure 2 together with gating strategies in other experiments.

**Statistics**. Values were expressed as mean ± SEM. Student's *t*-test was used to analyze difference between two groups and one-way analysis of variance (ANOVA) with Tukey's multiple comparison test was used to compare differences for more than two groups, except otherwise specified. Two-way ANOVA with Bonferroni post test was used to compare differences of tumor growth at different time points between groups. Log-rank (Mantel–Cox) test with Bonferroni correction was used to compare differences of survival between groups. *P*-value was calculated by PRISM software (GraphPad, San Diego, CA) and considered significant if it was < 0.05.

## Data availability

The data that support the findings of this study are available from the corresponding author upon request.

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

## Acknowledgements

This work is partly supported by the National Institutes of Health grants AI139473, AI107678, and DA033371 (to X.Y.C.). Microplate reader, BD FACSVerse, Applied Biosystems ViiA 7, Nikon Eclipse E600 microscope used in this work are supported by an Institutional Development Award (IDeA) from the National Institute of General Medical Sciences of the National Institutes of Health grant P20GM103430. We thank Lelia Noble at Rhode Island Hospital for her assistance in preparation and staining of paraffin sections.

## Author contribution

X.Y.C. designed and supervised the research. Y.C., X.Y.Z., and M.N.H. designed and conducted experiment. Y.C., X.Y.Z., M.N.H., and X.Y.C. analyzed data. P.K. and Y.W.Z. participated in discussion of this research. X.Y.C., Y.C., and X.Y.Z. wrote the manuscript.

## Additional information

**Competing interests:** The authors declare no competing interests.

