## [Peer Review File · Nature Communications]

Reviewers' comments:

Reviewer #1 (Remarks to the Author):

- 1) Very interesting, may solve the issue of adjuvant induced autoimmunity (Refer to it in your discussion).
- 2) Do the authors have a comparison of titers of antibodies against a peptide compared by HF to a regular adjuvant after 1st injection or a second.

Reviewer #2 (Remarks to the Author):

The authors reported for the first time use of radio frequency (RF) as a vaccine adjuvant (RFA), although a very similar observation has been reported in a patent application (WO2009144567) HIGH-ENERGY PULSED ELECTRIC FIELD VACCINE ADJUVANTS by a Russian group.

Compared to the previously explored laser adjuvant, RFA alone significantly augmented T-cell response, while lasers needed to be combined with other therapeutics such as microneedles (Wang J, PNAS, 2015; Wang J, Nat Commun, 2014).

This study suggests that the mechanism(s) of action of RFA may be similar to those of lasers; recruitment and activation of antigen presenting cells (APCs) by local and limited inflammation. Moreover, heat proved to play a significant role, which is consistent with a recent report (Ye Y, Sci Immunol, 2017). This report, however, failed to identify molecular cues, especially pathway(s) activated by RFA and/or subsequent heat generation to ultimately activate MyD88 and augment immune responses including CD8+T cell response.

In addition, more pieces of information on technical details of the RFA treatment, reproducibility of each experiment, and statistical strategy should be provided.

Major

1. In page 10, line 289: the authors state that "It has been reported that the majority of endogenous danger signals or damage-associated molecular patterns (DAMPs) can bind to toll-like receptors (TLRs)". Indeed, heat generation has been reported to increase the expression of heat shock protein, ROS generation, recruitment of immune cells, and blood and lymphatic flow, resulting in activation of innate and adaptive immune responses. However, none of these signals was examined in this study. As a result, mechanism(s) of action of the RFA remain unclear. In addition, in page 12, line 342: the authors state "To our knowledge, RFA represents the first physical adjuvant that simultaneously augments both arms of adaptive immunity with adjuvant potency". In order to confirm this conclusion, it is critical to specify pathway(s) involved in augmentation of both humoral and cell-mediated immunity.

2. Macrophages can express CD11c and MHC-II; contrary, dendritic cells can express F4/80. The prototypical markers used in Figure 2 to distinguish migratory DCs, lymphoid tissue-resident DCs and macrophages in skin-draining lymph nodes or skin are not adequate to draw any conclusions.

3. The authors state that there is no applicable side effect with the RFA, but, little evidence (ex, lack of systemic adverse effects) for this claim is presented in this study.

Minor

1. In page 3, line 34: "Alum adjuvant induces Th2-biased immune responses and enhances mainly humoral immune responses 2, 4" is not correct, as Alum induces Th1-Th2 mixed response in humans (Coffman RL, Sher A, Seder RA., Immunity, 2010)

2. Figure 1C-G: It is helpful to show the gating strategy to identify neutrophils, monocytes, macrophages, eosinophils, and skin mDCs.

3. In page 7, line 176: Ovalbumin from Sigma (Grade V) is known to contain LPS. The results and conclusion using this OVA may have been compromised by the existence of LPS, which strongly activates the innate immune response. If the authors perform LPS clean-up, it should be noted.

4. Figure 2, 3, 4, 5, 6: there is no description how many times these particular experiments were repeated to confirm reproducibility.
5. Figure 3: The authors describe "Log-rank test was used to compare differences in E.", but which method was used to correct for multiple comparisons between groups?
6. In Figure 6: The authors describe that "Student's t-test was used to compare differences between groups". In this experiment, there are three genotypes crossed with two treatments. Please clarify how the authors perform the statistical analysis. In Figure 6C: there are 3 test groups. Which method was used to correct for multiple comparisons between groups?
7. In Material and Method section, Page 14, Line 417: the authors state that "This device has three energy settings (low, medium, high) and high energy setting was used in this study". Please provide more specific pieces of information including frequency and actual power density used in this study. It is helpful to include detailed methodology how the authors determined these parameters on the mouse skin. In addition, technical details how the authors established and reproducibly perform the application help other investigators reproduce the results.
8. In page 12, line 342: the authors state that "To our knowledge, RFA represents the first physical adjuvant that simultaneously augments both arms of adaptive immunity with adjuvant potency". However, nanosecond pulsed laser is reported to augment both CD4 and CD8 T cell responses when it was combined with the intramuscular administration of a model vaccine (Chen X, PLOS ONE, 2010). Moreover, the same nanosecond pulsed laser alone was sufficient to enhance anti-tumor immunity when combined with dendritic cell vaccine (Chen X, Clin Cancer Res, 2012).

A point-by-point response to reviewers

First of all, we would like to thank reviewers for their valuable comments and suggestions. We have conducted new experiments and included new data in this revised manuscript, including (1) comparison of RFA with AddaVax to boost pandemic 2009 H1N1 influenza (pdm09) vaccination (Fig. 8), (2) evidence of heat shock protein 70 (HSP70) induction following RFA treatment (Fig. 9A & 9B), (3) local and systemic safety following pdm09 vaccination in the presence of RFA (Fig. S5), (4) effects of RFA on DC subsets in skin and draining lymph nodes (Fig. 2 & 3). Besides the requested experiments, we also conducted adoptive transfer of OT-I T cells to confirm activation of OVA-specific CD8+ T cells following OVA immunization in the presence of RFA (Fig. 6). We also thoroughly addressed other minor concerns raised by reviewers either in the following point-by-point response or in the body of the manuscript whenever appropriate. These suggestions and modifications have improved the manuscript significantly.

Reviewer #1 (Remarks to the Author):

1) Very interesting, may solve the issue of adjuvant induced autoimmunity (Refer to it in your discussion).

Response: We have discussed the less likelihood of RFA to induce autoimmune/inflammatory syndromes induced by adjuvant (ASIA) in the revised manuscript (P16, 1st paragraph).

2) Do the authors have a comparison of titers of antibodies against a peptides compared by HF to a regular adjuvant after 1st injection or a second.

Response: We compared RFA and MF59-like AddaVax adjuvant to boost pdm09 vaccine-induced HAI titer and protection against lethal viral challenges. We found RFA was at least comparable to AddaVax to boost pdm09 vaccination in murine models (Fig. 8).

Reviewer #2 (Remarks to the Author):

The authors reported for the first time use of radio frequency (RF) as a vaccine adjuvant (RFA), although a very similar observation has been reported in a patent application (WO2009144567) HIGH-ENERGY PULSED ELECTRIC FIELD VACCINE ADJUVANTS by a Russian group. Compared to the previously explored laser adjuvant, RFA alone significantly augmented T-cell response, while lasers needed to be combined with other therapeutics such as microneedles (Wang J, PNAS, 2015; Wang J, Nat Commun, 2014).

This study suggests that the mechanism(s) of action of RFA may be similar to those of lasers; recruitment and activation of antigen presenting cells (APCs) by local and limited inflammation. Moreover, heat proved to play a significant role, which is consistent with a recent report (Ye Y, Sci Immunol, 2017). This report, however, failed to identify molecular cues, especially pathway(s) activated by RFA and/or subsequent heat generation to ultimately activate MyD88 and augment immune responses including CD8+T cell response.

In additional, more pieces of information on technical details of the RFA treatment, reproducibility of each experiment, and statistical strategy should be provided.

Response: We thank this reviewer to provide closely related literature to our work. We have incorporated related literature in the discussion portion of the revised manuscript. To address the lack of molecular cues, we identified HSP70 as one endogenous danger signal stimulated by RFA, which likely mediated the observed RFA effects. In addition, we also provided more technical details about the RFA treatment, reproducibility of each experiments, and more information about the statistical analysis.

Major

1. In page 10, line 289: the authors state that "It has been reported that the majority of endogenous danger signals or damage-associated molecular patterns (DAMPs) can bind to toll-like receptors (TLRs)". Indeed, heat generation has been reported to increase the expression of heat shock protein, ROS generation, recruitment of immune cells, and blood and lymphatic flow, resulting in activation of innate and adaptive immune responses. However, none of these signals was examined in this study. As a result, mechanism(s) of action of the RFA remain unclear. In addition, in page 12, line 342: the authors state "To our knowledge, RFA represents the first physical adjuvant that simultaneously augments both arms of adaptive immunity with adjuvant potency". In order to confirm this conclusion, it is critical to specify pathway(s) involved in augmentation of both humoral and cell-mediated immunity.

Response: We thank the reviewer to raise this question. We identified HSP70 as one endogenous danger signal induced after RFA treatment (Fig. 9A & 9B). The available literature on the potent adjuvant effects of HSP70 is well in line with the observed RFA effects, hinting HSP70 may be the key endogenous danger signal mediating RFA effects (P16, 3rd paragraph).

2. Macrophages can express CD11c and MHC-II; contrary, dendritic cells can express F4/80. The prototypical markers used in Figure 2 to distinguish migratory DCs, lymphoid tissue-resident DCs and macrophages in skin-draining lymph nodes or skin are not adequate to draw any conclusions.

Response: We agree dendritic cells (DCs) cannot be simply gated based on CD11c, especial when different DC subsets exist in skin and draining lymph nodes. To address this, we stained skin and lymph node cells with antibodies against CD11c, MHC II, CD11b, Langerin, CD103, CD8 α , and CD80 in order to identify and explore RFA effects on different DC subsets in the revised manuscript (Fig. 2 & 3).

3. The authors state that there is no applicable side effect with the RFA, but, little evidence (ex, lack of systemic adverse effects) for this claim is presented in this study.

Response: We added local and systemic safety data in the revised manuscript (Fig. S5).

Minor

1. In page 3, line 34: "Alum adjuvant induces Th2-biased immune responses and enhances mainly humoral immune responses 2, 4" is not correct, as Alum induces Th1-Th2 mixed response in humans (Coffman RL, Sher A, Seder RA., Immunity, 2010)

Response: Alum adjuvant induces Th1 and Th2-mixed immune responses, which doesn't contradict with the description that 'Alum adjuvant induces Th2-biased immune responses' since Alum adjuvant more strongly induce Th2 versus Th1 immune responses (Petrovsky et al., Immunology and cell biology, 2004; Sunita et al., Front Immunol. 2013; James et al., J Immunol, 1999).

2. Figure 1C-G: It is helpful to show the gating strategy to identify neutrophils, monocytes, macrophages, eosinophils, and skin mDCs.

Response: We followed a published method for gating the different cell types. In brief, live cells were gated based on FSC and SSC profile and then plotted based on expression of CD11b and CD11c.

For gating neutrophils, CD11b⁺CD11c⁻ cells were gated and then plotted based on Ly6C and Ly6G. Ly6C⁺Ly6G^{hi} cells were gated and analyzed for F4/80 expression. F4/80⁻ cells were gated as neutrophils.

For gating eosinophils, CD11b⁺CD11c⁻ cells were gated and then plotted based on Ly6C and Ly6G. Ly6C⁺Ly6G^{int} cells were gated and analyzed for F4/80 expression. F4/80^{int} cells were gated as eosinophils.

For gating mDCs, CD11b⁺CD11c⁺ cells were gated and then plotted based on MHC II and F4/80. MHC II^{hi}F4/80^{low} cells were gated as mDCs.

For gating macrophages and inflammatory monocytes, CD11b⁺CD11c⁻ cells were gated and then analyzed for F4/80 expression. F4/80^{hi} cells were gated as macrophages. F4/80^{int} cells were gated and analyzed for Ly6C expression. Ly6C⁺ cells were gated as inflammatory monocytes.

3. In page 7, line 176: Ovalbumin from Sigma (Grade V) is known to contain LPS. The results and conclusion using this OVA may have been compromised by the existence of LPS, which strongly activates the innate immune response. If the authors perform LPS clean-up, it should be noted.

Response: Endotoxin levels of OVA were reduced with Detoxi-Gel Endotoxin Removing Column (20344, Thermo Fisher Scientific) from original ~60 EU/ml to less than 15 EU/ml determined by Pierce LAL Chromogenic Endotoxin Quantification Kit (88282, Thermo Fisher Scientific) for in vivo application (See Materials and Methods).

4. Figure 2, 3, 4, 5, 6: there is no description how many times these particular experiments were repeated to confirm reproducibility.

Response: Experiments in all figures were repeated at least once and representative results were shown.

5. Figure 3: The authors describe "Log-rank test was used to compare differences in E.", but which method was used to correct for multiple comparisons between groups?

Response: Log-rank test with Bonferroni correction was used to compare differences in E.

6. In Figure 6: The authors describe that "Student's t-test was used to compare differences between groups". In this experiment, there are three genotypes crossed with two treatments. Please clarify how the authors perform the statistical analysis. In Figure 6C: there are 3 test groups. Which method was used to correct for multiple comparisons between groups?

Response: We reanalyzed the multiple-group data using One-way ANOVA with Tukey's multiple comparison test in the revised manuscript (Fig. 9C & 9D, Fig. S7).

7. In Material and Method section, Page 14, Line 417: the authors state that "This device has three energy settings (low, medium, high) and high energy setting was used in this study". Please provide more specific pieces of information including frequency and actual power density used in this study. It is helpful to include detailed methodology how the authors determined these parameters on the mouse skin. In addition, technical details how the authors established and reproducibly perform the application help other investigators reproduce the results.

Response: This device works at around 1MHz frequency based on the manufacturer. We selected high energy level and 1.5 minutes of treatment time in order to induce significant thermal stress without skin damage for exploration of vaccine adjuvant effects. More technical details have been described in Materials and Methods.

8. In page 12, line 342: the authors state that "To our knowledge, RFA represents the first physical adjuvant that simultaneously augments both arms of adaptive immunity with adjuvant potency". However, nanosecond pulsed laser is reported to augment both CD4 and CD8 T cell responses when it was combined with the intramuscular administration of a model vaccine (Chen X, PLOS ONE, 2010). Moreover, the same nanosecond pulsed laser alone was sufficient to enhance anti-tumor immunity when combined with dendritic cell vaccine (Chen X, Clin Cancer Res, 2012).

Response: We revised the discussion and removed the claim that RFA represents the first physical adjuvant that simultaneously augment both arms of adaptive immunity.

Point by point response to Editor

In particular, we ask that you perform additional experiments to show that RF is at least as good as other adjuvants, as requested by Reviewer 1. We also ask that you provide further mechanistic insight into how RF affects the immune response, and address concerns regarding flow cytometry, as requested by Referee 2. Please address Referee 2's concerns relating to the need to strengthen the claim that RF has fewer side effects than other approaches, and for increased numbers of animal and strengthen statistical analyses. Finally, please also ensure that the revised manuscript includes more technical details on the use of RF, as requested by Referee 2.

Response: We have conducted new experiments to address these questions raised by reviewers. In brief, we conducted pdm09 vaccine immunization study to compare RFA with AddaVax adjuvant; we identified RFA could induce HSP70 release to potentially mediate RFA effects; we also included local and systemic safety data of RFA to boost pdm09 vaccination in the revised manuscript. We have also repeated experiments to verify our findings and strengthened statistical analyses. Lastly we also included more technical details on the use of RF in the revised manuscript.

Additionally, in their commentary to the editor the reviewer also made a number of points that additionally require addressing and have been included below for you to address:

“This is a novel aspect to deal with the disadvantages of adjuvants currently employed in vaccine (i.e. refer to the autoimmune diseases induced by adjuvants (ASIA).”

Response: We have discussed the safety advantage of RFA in the revised manuscript.

“Employing the HF tool to a 4 cm², an inflammatory reaction is induced which disappear in 24 h. The MyD88 and danger signals are activated (similarly to other adjuvants). The HF was more valuable than the chemical agents. The claim is that there are fewer adverse effects. Yet how long was the follow-up ? what are the human implications ? Should we HF longer follow up? How long? Which "strength"?”

Response: We included the local and systemic safety data on RFA to boost pdm09 vaccination (Fig. S5). Considering the major adverse reactions associated with RFA are skin overheating, if there were no skin reactions within 2 days after immunization, no skin reactions would occur later on. For systemic reactions, rectal temperature slightly increased at 6 hours and then returned to baseline levels at 24 hours in RFA and AddaVax groups.

“Do you have a real virus (i.e. Influenza) experiments? Titer of antibodies generated ?”

Response: We conduct pdm09 vaccine immunization study to compare RFA with MF59-like AddaVax adjuvants and found RFA was at least comparable to AddaVax adjuvant in boosting pdm09 vaccination (Fig. 8).

We therefore invite you to revise and resubmit your manuscript, taking into account the points raised. Please highlight all changes in the manuscript text file.

Response: The major changes have been highlight in red in the revised manuscript.

REVIEWERS' COMMENTS:

Reviewer #2 (Remarks to the Author):

The authors addressed concerns raised and the manuscript is acceptable. The graphic in Figure S7 needs to be improved.

REVIEWERS' COMMENTS:

Reviewer #2 (Remarks to the Author):

The authors addressed concerns raised and the manuscript is acceptable. The graphic in Figure S7 needs to be improved.

Response: The graphic in Supplementary Figure 8 (original Figure 7) has been improved as suggested.